

# Exome sequencing study of 20 patients with high myopia

Ling Wan[1,2], Boling Deng[2], Zhengzheng Wu[2] and Xiaoming Chen[1]

[1] Department of Ophthalmology, West China Hospital, Sichuan University, Chengdu, Sichuan, China
[2] Department of Ophthalmology, Sichuan Academy of Medical Sciences, Sichuan Provincial People's Hospital and Affiliated Hospital of University of Electronic Science and Technology, Chengdu, Sichuan, China

## ABSTRACT

**Background:** High myopia is a common ocular disease worldwide. To expand our current understanding of the genetic basis of high myopia, we carried out a whole exome sequencing (WES) study to identify potential causal gene mutations.

**Methods:** A total of 20 individuals with high myopia were exome sequenced. A novel filtering strategy combining phenotypes and functional impact of variants was applied to identify candidate genes by multi-step bioinformatics analyses. Network and enrichment analysis were employed to examine the biological pathways involved in the candidate genes.

**Results:** In 16 out of 20 patients, we identified 20 potential pathogenic gene variants for high myopia. A total of 18 variants were located in myopia-associated chromosomal regions. In addition to the novel mutations found in five known myopia genes (*ADAMTS18*, *CSMD1*, *P3H2*, *RPGR*, and *SLC39A5*), we also identified pathogenic variants in seven ocular disease genes (*ABCA4*, *CEP290*, *HSPG2*, *PCDH15*, *SAG*, *SEMA4A*, and *USH2A*) as novel candidate genes. The biological processes associated with vision were significantly enriched in our candidate genes, including visual perception, photoreceptor cell maintenance, retinoid metabolic process, and cellular response to zinc ion starvation.

**Discussion:** Systematic mutation analysis of candidate genes was performed using WES data, functional interaction (FI) network, Gene Ontology and pathway enrichment. FI network analysis revealed important network modules and regulator linker genes (*EP300*, *CTNNB1*) potentially related to high myopia development. Our study expanded the list of candidate genes associated with high myopia, which increased the genetic screening performance and provided implications for future studies on the molecular genetics of myopia.

Corresponding author
Xiaoming Chen,
chenxm058@163.com

## INTRODUCTION

Myopia, recognized as the most common ocular disease (*Holden et al., 2016*; *Shi et al., 2011*; *Vitale, Sperduto & Ferris, 2009*), has a prevalence in 20–30% of Australian, American, and Western European populations (*Kempen et al., 2004*). One extreme type of
myopia is high myopia, defined as a refractive error of at least −6.0 diopters (D) or axial length >26 mm (*Young et al., 1998*). It occurred in about 2.7% of the world population in 2000 and has shown significant increase in prevalence over the last two decades (*Holden et al., 2016*). Although myopia is usually a benign disorder that can be corrected by glasses and contact lenses, patients with high myopia are at increased risk of other complications, such as cataracts, glaucoma, retinal detachment, and posterior staphyloma (*Morgan, Ohno-Matsui & Saw, 2012*; *Saw et al., 2005*; *Xu et al., 2006*).

Genetic factors play a critical role in the development of high myopia based on family aggregation and twin studies (*Hammond et al., 2001*; *Katz, Tielsch & Sommer, 1997*), with its heritability estimated to be over 70% in large twin studies (*Dirani et al., 2006*; *Lopes et al., 2009*). Family linkage studies have identified 26 Quantitative Trait Loci (QTLs) (OMIM: 160700) so far (*Andrew et al., 2008*; *Li et al., 2009*; *Nakanishi et al., 2009*; *Schwartz, Haim & Skarsholm, 1990*; *Young et al., 1998*). However, only seven QTLs are fine-mapped to the causal genes (Table S1), a large proportion of linkage signal found in family studies still remain to be elucidated.

The inheritance mode for familial form of high myopia is complex. It may be inherited in an autosomal recessive, autosomal dominant, or X-linked recessive manner (*Ng et al., 2009*; *Zhang, 2015*). Quite a few disease-associated genes have been identified in individual studies, such as dominant genes *SCO2* (*Tran-Viet et al., 2013*), *ZNF644* (*Shi et al., 2011*), and *P4HA2* (*Guo et al., 2015*); recessive genes *LRPAP1* (*Aldahmesh et al., 2013*) and *LEPREL1* (*Mordechai et al., 2011*); X-linked gene *ARR3* (*Xiao et al., 2016*). As the sequencing costs keep dropping, whole exome sequencing (WES) has been gradually adopted in studies of familial or early-onset high myopia. Variants in *LRPAP1*, *CTSH*, *LEPREL1*, *ZNF644*, *SLC39A5*, and *SCO2* genes were comprehensively screened in 298 families with early-onset high myopia (*Jiang et al., 2014*). However, only a small proportion of patients were found to have deleterious mutations in these six genes. *Sun et al. (2015)* investigated mutations in 234 genes associated with retinal dystrophies in 298 patients with early-onset high myopia. They found that 34 of 234 genes had potential pathogenic mutations including *GNAT1*, *GUCY2D*, *COL2A1*, *COL11A1*, *PRPH2*, *FBN1*, *TSPAN12*, *CACNA1F*, *OPA1*, *PAX2,* and *RPGR*. Recently, *Kloss et al. (2017)* analyzed 14 families of high myopia through WES combining with QTL overlapping analysis and identified 73 rare and 31 novel pathogenic variant candidates. *Jin et al. (2017)* took another approach searching for de novo mutations through WES in 18 family trios with healthy parents and early-onset high myopia children. In addition to the novel pathogenic gene *BSG*, two known high myopia candidate genes (*LEPREL1* and *GRM6*), three oculopathy-related genes (*FAM161A*, *GLA*, and *CACNA1F*), and a further possible gene (*MAOA*) were also identified in this trio study.

The high prevalence of high myopia in East Asian population (*Holden et al., 2016*; *Wu et al., 2015*) and previous WES studies suggested the complex genetic mechanisms behind the disease despite its strong heritability in families. In this study, we applied a novel phenotype driven variant filtering approach on the WES data of 20 patients with familial high myopia to search for potential causal mutations. We also explored the functions of the candidate genes through gene expression profiling, network and pathway enrichment analysis.

## MATERIALS AND METHODS

### Human subjects

This study included a total of 20 samples from 19 families (nine males and 11 females) of Han Chinese ancestry with non-syndromic familial high myopia. The human subjects were recruited at the ophthalmic clinic at Sichuan Academy of Medical Sciences & Sichuan Provincial People's Hospital, Chengdu, China. Approval for the study was provided by the Institutional Review Board of Sichuan Provincial People's Hospital. Written informed consent forms were signed by all patients before the collection of peripheral blood and clinical data. The refractive errors of all patients were greater than −6.00 D, as diagnosed by a team of high myopia specialists.

### Exome sequencing and variant calling

Genomic DNA was isolated from peripheral blood samples of patients by using Genomic DNA Extraction Kit (Invitrogen, South San Francisco, CA, USA). The sequencing libraries were prepared and captured using SureSelect Human All Exon V6 kit (Agilent Technologies, Santa Clara, CA, USA) following the manufacturer's instructions. Paired-end (2 × 150 bp) NGS was performed using HiSeq X-10 (Illumina, San Diego, CA, USA) according to the manufacturer's protocol.

Raw sequencing reads were filtered using the Trimmomatic program (http://www.usadellab.org/cms/index.php?page=trimmomatic) to remove reads with low sequencing quality at both ends. The cleaned reads were mapped to human reference genome (GRCH37) by the BWA-MEM software (*Li & Durbin, 2009*), and Polymerase Chain Reaction (PCR) duplications were removed by Picards (http://broadinstitute.github.io/picard). Variant calling was performed by a consensus call method to reach a good balance of high sensitivity and low false positive rate. Only variants detected by at least two methods out of four haplotype-based calling algorithms (Platypus, samtools, freebayes, and GATK haplotype caller) were kept in the final variant file. Variant annotation including gene functional consequence (gene context, amino acid change, splicing effect etc.), pathogenicity predictions by multiple computational methods (SIFT, Polyphen2, MutationTaster, CADD etc.), as well as healthy population allele frequencies from 1000G (*Sudmant et al., 2015*), ExAC (http://exac.broadinstitute.org/), and ESP (http://evs.gs.washington.edu/) database (*Lek et al., 2016*) were performed using the VEP software (*McLaren et al., 2016*).

### Variant filtering

To help narrow down the potential candidate variants for high myopia, we took a phenotype driven strategy focusing on rare variants in ocular diseases and high myopia phenotype related genes as illustrated in Fig. 1. First, we used a maximum allele frequency of 0.005 in any of the three public population databases: 1000G (http://browser.1000genomes.org/index.html), ExAC, and ESP dataset to filter out common variants. Then, only variants with high functional impact (loss-of-function, or computationally predicted pathogenic missense variants) were considered. Pathogenicity of missense mutations was assumed if predicted pathogenic by at least six out of eight computational methods (SIFT, PolyPhen2, LRT, MutatationTaster,

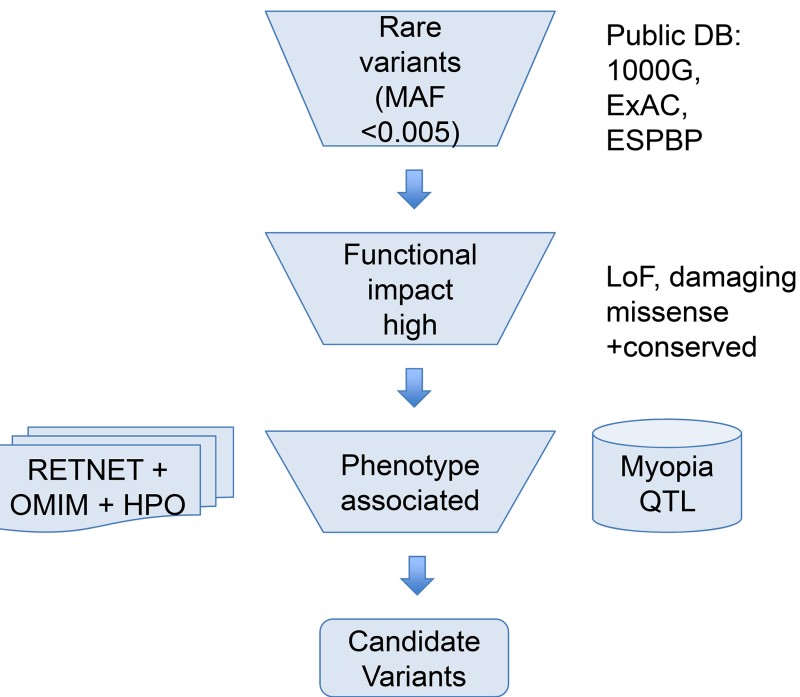

**Figure 1 Overview of the variant filtering strategy and analysis of WES data.** We applied a phenotype driven strategy to filter the potential candidate variants for high myopia. Variant frequency in populations, mutation functional impact, and phenotype associations from RetNet, HPO, and OMIM are combined to narrow down the candidate variants.

MutationAssessor, FATHMM, CADD, DANN). Lastly, we focused on phenotype relevant genes. We considered only genes meeting both of the following two criteria. Firstly, genes must be associated with eye-related diseases. The ocular disease genes were assembled from the following three databases: genes associated with the myopia phenotype term (HP:0000545) in Human Phenotype Ontology (HPO: http://human-phenotype-ontology.github.io), or ocular disease genes from RetNet database (http://www.sph.uth.tmc.edu/retnet/) and OMIM (https://www.omim.org/). Secondly, genes must be known high myopia genes from previous publications (*Aldahmesh et al., 2013*; *Chen et al., 2013*; *Jiang et al., 2014*; *Jin et al., 2017*; *Kloss et al., 2017*; *Shi et al., 2011*; *Sun et al., 2015*; *Tran-Viet et al., 2013*) (complete gene list of known genes and ocular disease genes in Table S2) or within the myopia-associated (MYP) QTL regions (Table S1). Because not all patients have syndromic severe ocular diseases, a few variants in genes causing dominant OMIM ocular diseases not satisfying clinical phenotypes were excluded.

## Functional enrichment and network analysis

We chose a recent RNA-seq expression profile GSE94437 and a gene microarray expression profile GSE41102 to explore the expression pattern of the compiled complete list of ocular disease genes. For the GSE94437, RPKM was transformed by $\log_2(RPKM + 0.001)$. To test the expression differences of the 709 ocular disease genes versus other genes, one-sided $t$-test was applied to obtain the statistical significance. The Gene Ontology (GO) is a hierarchically organized, controlled vocabulary to

consistently describe and annotate gene products (*Ashburner et al., 2000*). Joint terms may give insight on the shared biological processes, and enrichment analysis can make use of term–term relationships (*Huang, Sherman & Lempicki, 2009*; *Kuleshov et al., 2016*). The ReactomeFIViz app (*Wu et al., 2014*) was designed to find GO, pathways and network patterns. Thus we used the ReactomeFIViz app and Cytoscape 3.5.1 to perform GO enrichment analysis of candidate genes. Using all the protein-coding genes in the genome as background, we carried out the GO enrichment for both the 12 candidate gene set and the 709 eye disease gene set. ReactomeFIViz was also used to explore the network features of candidate genes in the manually curated pathway-based protein functional interaction (FI) network covering over 60% of human proteins (*Wu, Feng & Stein, 2010*).

## RESULTS

In this study, WES from 20 patients with high myopia were performed to find disease-associated mutations, followed by pathway, GO enrichment and network analysis (Materials and Methods). From WES, we generated an average of 10.7 Gb of sequence with $121\times$ mean target coverage for each individual as paired-end, 150 bp reads. All samples had more than 95% of target bases covered by at least 10 reads. There are an average of 49,904 Single Nucleotide Polymorphisms (SNPs) and small indels called per sample. After multiple steps of variant filtering considering Minor Allele Frequency (MAF), functional impact, phenotype associations and QTL overlapping information, a total of 20 potential pathogenic (three splicing, one frame-shift, and 16 deleterious missense) heterozygous variants in 12 genes were identified in 16 patients (Table 1).

There were eight pathogenic variants identified from five known high myopia genes, including *ADAMTS18*, *CSMD1*, *P3H2*, *RPGR* and *SLC39A5*. Patient R0020 had a clear molecular diagnosis with one missense variant rs756666376 in gene *SLC39A5* (MYP24, autosomal dominant inheritance). Rs756666376 (p.Gly293Arg) is extremely rare in healthy populations, absent in 1000G data, with only one heterozygote in 4,325 East Asians (MAF = 0.00012) and two heterozygotes in 33,359 Non-Finnish Europeans (MAF = 0.00003) in the ExAC database. The SNP is highly conserved and predicted to be deleterious by all of the eight computational methods in our variant annotation. The *RPGR* gene is located on the Xp21.1 and plays a role in ciliogenesis and photoreceptor integrity. In gene *RPGR*, sample R0015 had a heterozygous loss-of-function mutation in the second intron (c.154+4A>G) potentially affecting splicing as predicted by the adaptive boosting method in the splicing consensus region (*Jian, Boerwinkle & Liu, 2014*) and sample R0016 had a rare heterozygous pathogenic missense mutation (rs774982456, c.1832A>G).

Other than the known high myopia genes, we also discovered 12 rare potentially pathogenic heterozygous variants located in other MYP QTL regions, suggesting potential new candidate genes explaining these QTLs. Among these ocular disease related or direct high myopia related genes, five genes (*CSMD1*, *HSPG2*, *RPGR*, *SEMA4A* and *USH2A*) have pathogenic variants in multiple patients.

To look deep into the genes with pathogenic variants associated with high myopia, we assembled a list of 709 genes associated with eye diseases (details in variant filtering

**Table 1  Variants identified in patients with high myopia.**

| Gene | Inheritance | Type | Patient ID | Position | Variations Nucleotide | Variations Amino acid | dbSNP | QTL |
|---|---|---|---|---|---|---|---|---|
| ABCA4 | AR | Ocular disease gene | R0025 | 1:94490591 | c.4553G>A | p.Ser1518Asn | – | MYP14 |
| ADAMTS18 | AR | Known HM gene | R0017 | 16:77334288 | c.2546G>A | p.Gly849Asp | – | – |
| CEP290 | AR | Ocular disease gene | R0031 | 12:88519100 | c.1112T>C | p.Val371Ala | – | MYP3 |
| | | | R0026 | 8:2975943 | c.6408C>G | p.Asn2136Lys | rs778164827 | MYP10 |
| CSMD1 | | Known HM gene | R0027 | 8:3165341 | c.3826G>A | p.Glu1276Lys | rs534926586 | MYP10 |
| | | | R0029 | 8:3165341 | c.3826G>A | p.Glu1276Lys | rs534926586 | MYP10 |
| HSPG2 | AR | Ocular disease gene | R0023 | 1:22154919 | c.12238G>A | p.Val4080Met | – | MYP14 |
| | | | R0011 | 1:22207207 | c.1940G>A | p.Arg647Gln | – | MYP14 |
| P3H2 | AR | Known HM gene | R0031 | 3:189692463 | c.1336C>T | p.Leu446Phe | – | – |
| PCDH15 | AR | Ocular disease gene | R0018 | 10:55973726 | c.1081_1082delGA | p.Asp361LeufsTer6 | – | MYP15 |
| RPGR | X-linked | Known HM gene | R0016 | X:38147035 | c.1832A>G | p.Asn611Ser | rs774982456 | MYP13 |
| | | | R0015 | X:38182648 | c.154+4A>G | Splicing | rs764483977 | MYP13 |
| SAG | AR | Ocular disease gene | R0021 | 2:234217902 | c.72_75+15delATCGGTGAGTGGTGCACAA | Splicing | rs771810575 | MYP12 |
| SEMA4A | AR/AD | Ocular disease gene | R0019 | 1:156146474 | c.1972C>T | p.Arg658Trp | rs756847201 | MYP14 |
| | | | R0012 | 1:156126310 | c.245A>G | p.Glu82Gly | – | MYP14 |
| SLC39A5 | AD | Known HM gene | R0020 | 12:56629416 | c.877G>A | p.Gly293Arg | rs756666376 | MYP24 |
| | | | R0018 | 1:216419965 | c.2771C>T | p.Pro924Leu | – | MYP14 |
| USH2A | AR | Ocular disease gene | R0019 | 1:216372968 | c.3811+1G>A | Splicing | – | MYP14 |
| | | | R0014 | 1:215821999 | c.14453C>T | p.Pro4818Leu | rs143344549 | MYP14 |
| | | | R0015 | 1:216498726 | c.1064T>C | p.Val355Ala | rs746683099 | MYP14 |

**Notes:**
Chromosome position in accordance with GRCh37/hg19 assembly.
The HM is abbreviation of the high myopia.

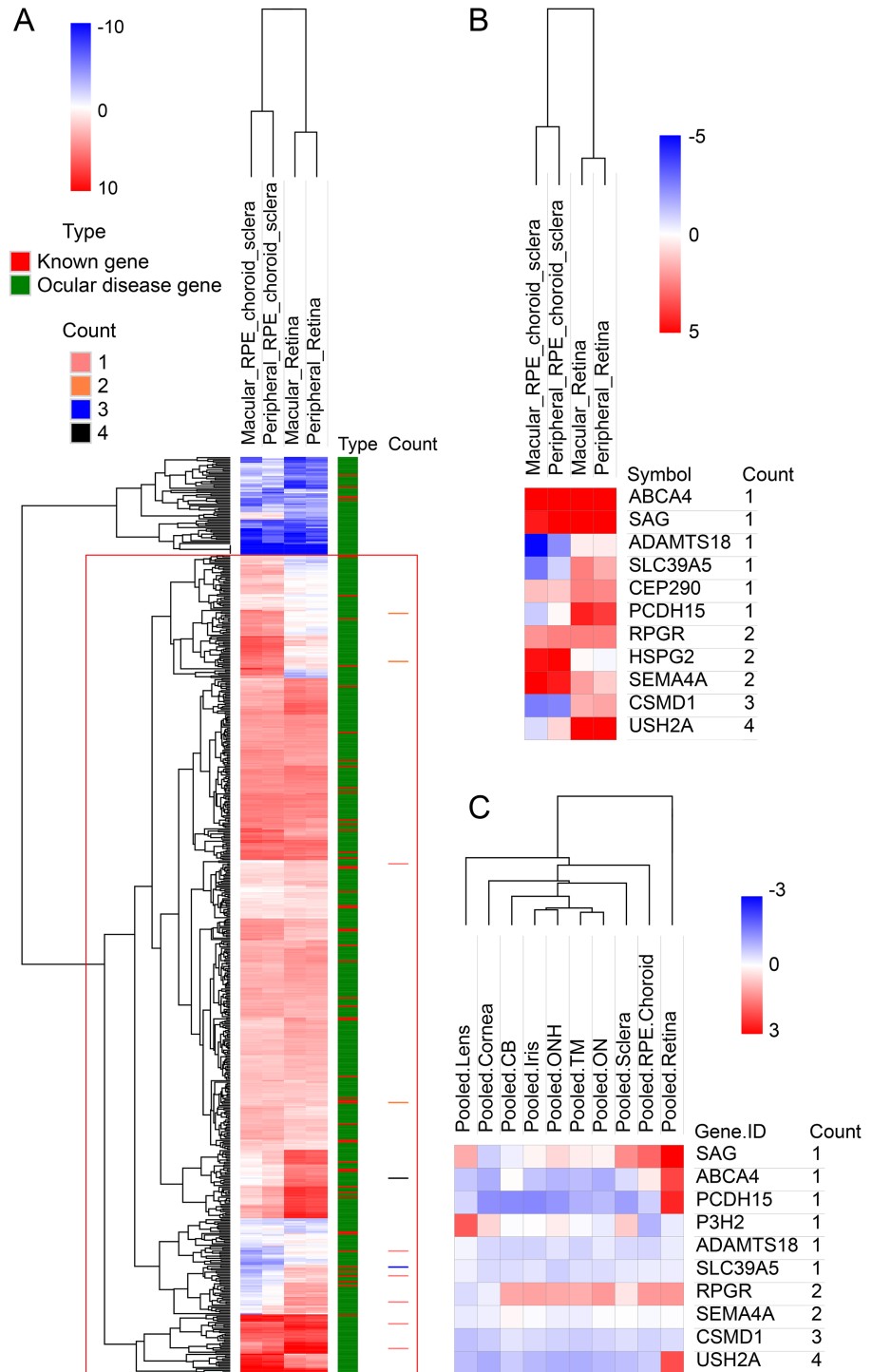

**Figure 2 Hierarchical cluster analysis of gene expression.** (A) The heatmap of 709 known genes associated with eye diseases in RNA-seq expression profile. The 709 genes were divided into known high myopia genes and ocular disease genes, as shown in different colors in type column. Count column represents the sample count of pathogenic variants for our candidate genes. (B) The heatmap of candidate genes in RNA-seq expression profile. (C) The heatmap of candidate genes in microarray expression profile.
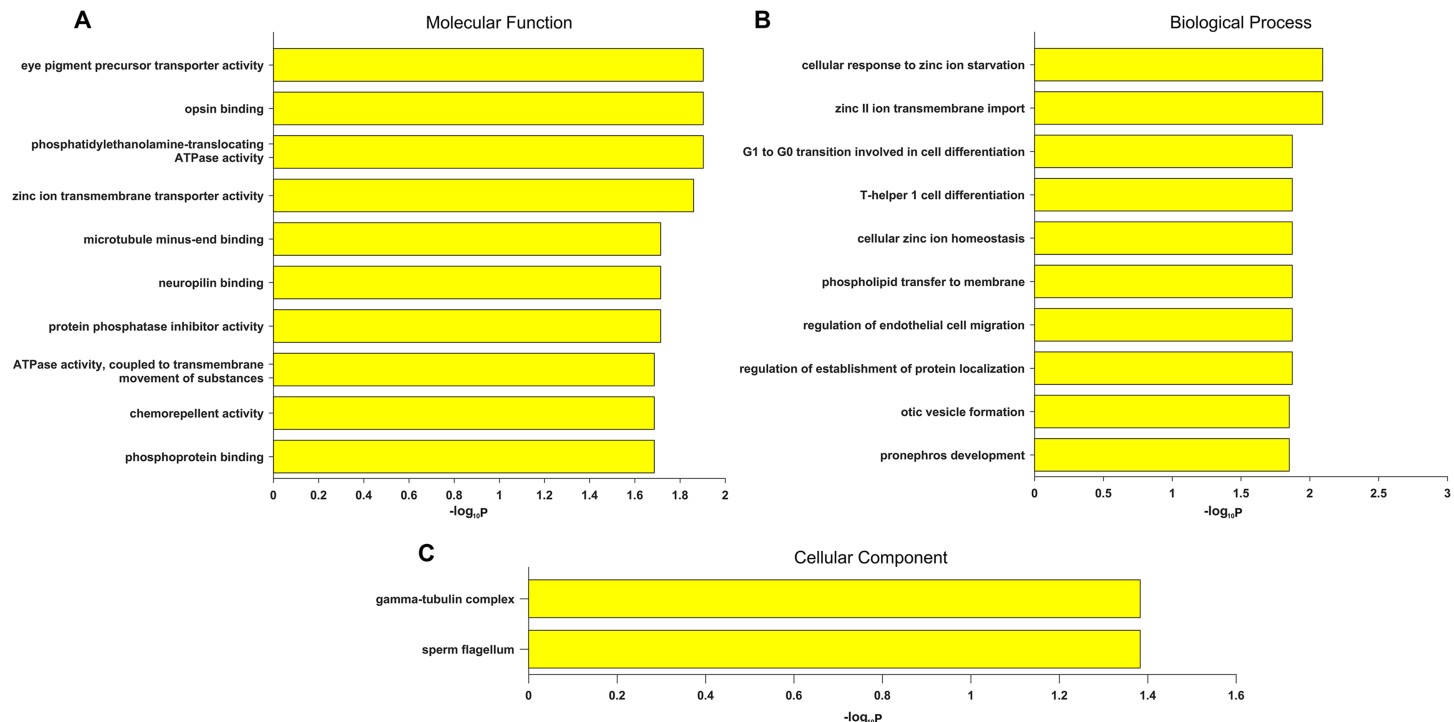

**Figure 3 Top 10 significantly enriched GO terms only in candidate genes and not in the 709 eye disease genes.** (A) Molecular function. (B) Biological process. (C) Cellular component. $x$-axis is the $-\log_{10}$ (BH adjusted $P$-value) from hypergeometric test of enrichment analysis. Complete list of enriched GO terms in Table S3.

section of Materials and Methods), including 58 known high myopia genes and 651 ocular disease genes (Table S2). Clinical phenotype related studies have shown that retinal, choroidal, and scleral lesions are high-frequency phenotypes in high myopia patients (*Brussee et al., 2014*; *Gupta et al., 2015*, *2017*). It would be natural to assume that myopia related genes would be highly expressed in retina and sclera tissues. To test this hypothesis, we first explored the expression pattern of all the 709 eye disease genes. The results showed that in the RNA-seq dataset, the expression of 709 genes was significantly higher than that of other genes in retina, retinal pigment epithelium, choroid and sclera ($p < 0.001$; Fig. S1). All candidate genes in our own high myopia cohort were located in the positive expression class when using the 709 eye disease genes as background (Fig. 2A). Looking further into the candidate gene expression differences among the three tissues, we found that five genes (*PCDH15*, *CEP290*, *SLC39A5*, *CSMD1* and *USH2A*) have much higher expression in retinas (macular retina and peripheral retina), while *ABCA4*, *SAG*, *RPGR*, and *SEMA4A* are highly expressed in both sclera and retina (Fig. 2B). Microarray dataset confirmed the higher expression in retina for genes *PCDH15* and *USH2A* (Fig. 2C).

Gene Ontology term enrichment was carried out for a comprehensive functional analysis of the 12 genes found in this study. Because our bioinformatics pipeline has a filtering step to include only eye disease related genes, this could lead to potential bias in the enrichment signal. Thus we did two GO enrichment analyses using either our 12 genes,

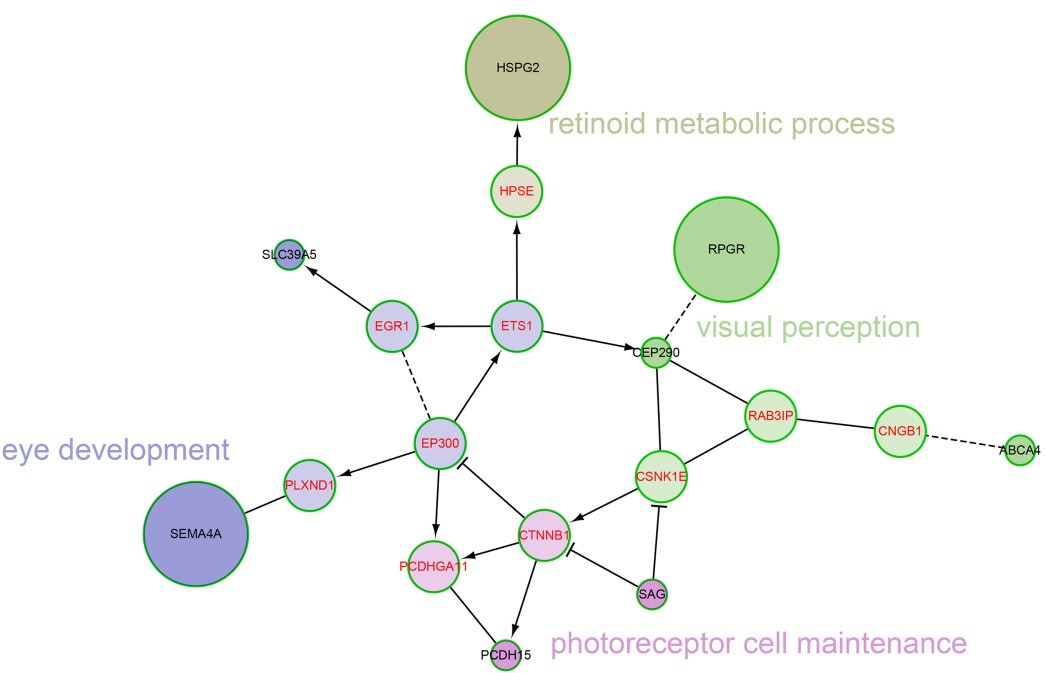

**Figure 4 The network modules associated with high myopia candidate genes and linker genes in the functional interaction (FI) network.** This network can be divided into four network modules. The background colors of genes represent different modules. Principal biological process of each module was described. The candidate genes are black in text color, and the linker genes are red in text color. The circle size of each candidate gene is proportional to the number of samples with pathogenic variants for the gene.

or the 709 eye disease genes (complete list of enriched GO terms from both analyses are in Table S3). Other than the enriched GO terms like visual perception, photoreceptor cell maintenance, retinoid metabolic process shared with the 709 genes, significant enrichment was uniquely observed in our candidate genes in eye pigment precursor transporter activity, opsin binding, zinc ion transmembrane related processes, microtubule binding etc. (Fig. 3). Interestingly, the visual phototransduction pathway was enriched by pathway analysis. Visual phototransduction is the photochemical reaction that takes place when light is converted to an electric signal in the retina (*Smith, Sivaprasad & Chong, 2016*). It's not surprising the malfunction of genes in this pathway will potentially lead to myopia. This pathway was also found highly enriched in differentially expressed genes during myopia development in a mouse study (*Metlapally et al., 2016*), which corroborated these genes' potential causal roles in high myopia.

We also explored the network characteristics in a FI network (*Wu, Feng & Stein, 2010*). Focused on the 12 genes from our own cohort, except for the direct interaction between *RPGR* and *CEP290*, other genes are scattered, but all genes can be linked to form a large network by some linker genes, such as *EP300*, *CTNNB1* (Fig. 4). A network clustering algorithm (*Newman, 2006*) divided this network into four network modules. The principal biological process of each module was eye development, retinoid metabolic process, visual perception and photoreceptor cell maintenance, respectively. There were some differences between modules in biological processes, but they were all related to

vision. When using the 709 genes as background, only GO term photoreceptor cell cilium was found enriched in the modules (Fig. S2). *EP300* encodes protein p300, which is a transcriptional co-activator regulating hundreds of genes' transcription via chromatin remodeling, and is important in the processes of cell proliferation and differentiation. Mutations in *EP300* are known to cause Rubinstein–Taybi syndrome (RTS). Interestingly, RTS patients with *EP300* mutations were frequently documented with severe myopia phenotype (*Bartholdi et al., 2007*; *Fergelot et al., 2016*). Various visual defects (including familial exudative vitreoretinopathy, retina detachment and myopia etc.) have also been reported in recent WES studies of people with mutations in *CTNNB1* (*Li et al., 2017*; *Panagiotou et al., 2017*).

## DISCUSSION

Whole genome sequencing and WES have now entered medical practice (*Biesecker & Green, 2014*). Whole genome sequencing is the most powerful method of disease gene identification, because this method targets both coding and regulatory non-coding variants, and can identify disease genes caused by SNVs, large indels, and other large structure variants as well. However, its cost is still daunting. Some previous studies have estimated that ~85% of Mendelian disease mutations are located within the coding region, canonical splice acceptor and donor sites (*Stenson et al., 2017*). Thus the WES becomes the sweet spot rather than Whole Genome Sequencing (WGS) (*Teer & Mullikin, 2010*). We used WES to identify causal gene mutations in 20 patients with high myopia. Potential pathogenic mutations were identified in 16 patients (80.0%, 16/20). A total of 20 mutations in 12 genes were potentially pathogenic variants for high myopia. Compared to the discovery rate in previous studies (*Chen et al., 2013*; *Kloss et al., 2017*; *Sun et al., 2015*), our phenotype driven filtering method yielded higher diagnostic rate.

In our study, *SEMA4A*, *RPGR,* and *HSPG2* each had two pathogenic mutations detected in our probands. Gene *SEMA4A* encodes a member of the semaphorin family of soluble and transmembrane proteins. This gene maps to chromosome 1p36, which is located within the MYP14 locus (*Abid et al., 2006*; *Wojciechowski et al., 2006*). Mutation in codon 345 from G to C and in codon 350 from T to G in exon 10 of the *SEMA4A* gene resulted in an asp345-to-his (D345H) and a phe350-to-cys (F350C) substitution, respectively. A link between those two mutations and retinitis pigmentosa and retinal degeneration has been demonstrated (*Abid et al., 2006*; *Berger, Kloeckener-Gruissem & Neidhardt, 2010*). Retinitis pigmentosa GTPase regulator (*RPGR*) is one of the main genes causing X-linked retinitis pigmentosa (XLRP). It was suggested that over 70% of XLRP cases were caused by *RPGR* mutations (*Breuer et al., 2002*; *Sharon et al., 2003*). Multiple studies have shown that female heterozygous *RPGR* mutation carriers had early-onset high myopia in one of two eyes, especially those with protein truncating variants (*Jin et al., 2006*; *Yokoyama et al., 2001*). Some families showed even full penetrance of high myopia in heterozygous carriers (*Parmeggiani et al., 2016*). Both patients with a RPGR mutation are female carriers in our study. The incomplete penetrance of XLRP phenotype of other RPGR carriers suggested that annual follow up examinations checking symptoms of Retinitis Pigmentosa (RP) is warranted. Gene *HSPG2*

encodes a protein called perlecan (*Farach-Carson & Carson, 2007*; *Warren et al., 2015*). Perlecan is a heparan sulfate proteoglycan, which interacts with many other proteins and has a variety of functions, such that mutation of the gene has pleiotropic effects. Schwartz–Jampel syndrome is a rare autosomal recessive skeletal dysplasia with myotonia, short stature, and low-set ears and myopia (*Kubrey, Solanki & Agrawal, 2015*). The two damaging missense mutations (c.12238G>A and c.1940G>A) in *HSPG2* each occurred in one patient. In our current cohort, four heterozygote variants in gene *USH2A* were identified, which fall within QTL MYP14. *USH2A* is a large gene with 73 exons and encodes at least two different isoforms. The mutations in the *USH2A* genes are responsible for 5~10% of the cases with retinitis pigmentosa and 60~90% of the cases with Usher syndrome type II (*Baux et al., 2014*; *Pennings et al., 2004*; *Slijkerman et al., 2016*).

Gene *ABCA4* is specifically expressed in cone and rod photoreceptor outer segments (*Molday, Rabin & Molday, 2000*). The ABCA4 protein is active following phototransduction. *ABCA4* mutations can result in multiple vision related phenotypes (*Cideciyan et al., 2009*; *Klevering et al., 2004*; *Shroyer et al., 1999*; *Valverde et al., 2007*) including Retinitis pigmentosa, Fundus flavimaculatus, Cone-rod dystrophy, and Stargardt disease (*Lin et al., 2016*), which is characterized by juvenile macular degeneration. In addition, a missense variant c.1268A>G in *ABCA4* was recently found to be responsible for myopia (*D'Angelo et al., 2017*). A novel missense rare variant (not observed in 1000G, ESP, or ExAC database) c.4553G>A (p.Ser1518Asn) was found in our patient R0025. *Yzer et al. (2012)* made a mutation analysis for gene *CEP290* and found that patients with a different mutation c.5587–1G>C showed myopia. Based on linkage and haplotype analysis, *Nallasamy et al. (2007)* identified a presumptive myopia locus in gene *PCDH15*. A novel frameshift mutation (c.1081_1082delGA) was found in one patient R0018. These functional evidences suggested that carriers of high impact variants in these severe recessive inheritance ocular disease genes might be at higher risk for high myopia. When we relaxed the computational prediction filtering criteria for missense variants to at least five or four out of eight methods, we obtained one or two extra candidate variants (Table S4). This suggested that our pipeline is relatively robust to the computational prediction cut-offs. Despite published evidences of the connections with myopia for the genes found in our cohort, none of the candidate variants were found in previous published sequencing studies of myopia (*Aldahmesh et al., 2013*; *Chen et al., 2013*; *Jiang et al., 2014*; *Jin et al., 2017*; *Kloss et al., 2017*; *Shi et al., 2011*; *Sun et al., 2015*; *Tran-Viet et al., 2013*), which highlighted the strong heterogeneity even in familial high myopia. The molecular functional impact of these mutations on the penetrance and severity of various ocular disease phenotypes would need further functional studies.

There are some limitations to our current study due to the small sample size and availability of family genetic data. Although our phenotype driven analysis approach has boosted the probabilities of the final variants associated with high myopia, the statistical significance of any genuine associated genes would need a large sample size gene-level burden test (*Lee et al., 2014*). Alternatively, extra genetic validation of the causal relationship of these variants with high myopia would require family segregation

analysis with extended family phenotyping and genotyping. In our study, R0027 (daughter) and R0029 (father) are the only two samples from the same family. After our bioinformatics filtering process, there are two candidate variants left in both patients and both variants are shared and consistent with Mendelian inheritance. One is the heterozygote c.3826G>A SNP in *CSMD1* listed in Table 1, the other is a missense variant c.2338C>G (p.Pro780Ala) in *COL4A5* on chromosome X with the daughter's genotype being a heterozygote and the father's genotype being a hemizygote. Deletions or pathogenic missense variants in *COL4A5* are known to cause X-linked dominant Alport syndrome (*Knebelmann et al., 1996*), however, neither patient exhibits the clinical phenotypes of Alport syndrome. Therefore, this *COL4A5* variant was excluded from our final candidate list.

## CONCLUSIONS

There is a large proportion of genetic heritability of high myopia still unexplained by known myopia genes. WES enables thorough and unbiased genetic analysis of candidate genes as well as novel gene discoveries. In this study, we took a novel bioinformatics screening approach combining ocular disease gene annotation, myopia phenotype to gene association, and rare variant functional effect filtering narrowing down potential causal variants. Systematic mutation analysis of high myopia genes was further analyzed by FI network, GO and pathway enrichment, which expanded our current understanding of high myopia. These variants, especially those on novel high myopia genes, expanded the mutation spectrum of myopia genes and provided clues for further genetic screening targets.

## ACKNOWLEDGEMENTS

We would like to thank all the patients for participating in this study, and Dr. Yongcheng Dong and Xin Ma for feedback on bioinformatics analysis.

### Funding

This work was supported by Specialized Research Fund for the Doctoral Program of Higher Education (Grant No. 20130181110079) and the National Major Scientific Equipment program (Grant No. 2012YQ12008005). The funders had no role in study design, data collection and analysis, decision to publish, or preparation of the manuscript.

### Grant Disclosures

The following grant information was disclosed by the authors:
Specialized Research Fund for the Doctoral Program of Higher Education: 20130181110079.
National Major Scientific Equipment program: 2012YQ12008005.

### Competing Interests

The authors declare that they have no competing interests.

## Author Contributions

- Ling Wan conceived and designed the experiments, performed the experiments, analyzed the data, contributed reagents/materials/analysis tools, prepared figures and/or tables, authored or reviewed drafts of the paper, approved the final draft.
- Boling Deng performed the experiments, approved the final draft.
- Zhengzheng Wu performed the experiments, approved the final draft.
- Xiaoming Chen conceived and designed the experiments, contributed reagents/materials/ analysis tools, authored or reviewed drafts of the paper, approved the final draft.

## Human Ethics

The following information was supplied relating to ethical approvals (i.e., approving body and any reference numbers):

Approval for the study was provided by the Institutional Review Board of Sichuan Provincial People's Hospital.

## Data Availability

The raw sequence data reported in this paper are deposited in the Genome Sequence Archive (Genomics, Proteomics & Bioinformatics 2017), Beijing Institute of Genomics (BIG), Chinese Academy of Sciences, under accession numbers CRA000764 and CRA000769, publicly accessible at http://bigd.big.ac.cn/gsa.

## Supplemental Information

Supplemental information for this article can be found online at http://dx.doi.org/10.7717/peerj.5552#supplemental-information.

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
