# Peer review of "Exome sequencing study of 20 patients with high myopia"

_PeerJ, doi:10.7717/peerj.5552_

## Round 0.1 · original submission · Major Revisions

· Academic Editor

Major Revisions

All the reviewers agree this is an interesting and clearly written study. Meanwhile they raise serious concern on causality claim and calls for more rigorous study. I suggest the authors to perform additional computation and add necessary discussions to response those comments.

Reviewer 1 ·

Basic reporting

This paper is well written and is easy to follow. In terms of writing, one minor suggestion is that the authors should provide some statistics or references to support their claim as "The high prevalence of high myopia in East Asian population" in the Introduction section.

Experimental design

The objective of this study is well defined and of high value. The experimental design was properly demonstrated and the proposed variant analysis approach was reasonable. The authors focus on rare variants uncovered by exome sequencing, so it is helpful to discuss the effect of common variants on high myopia and if any GWAS have been conducted to study this disease.

Validity of the findings

Several potential disease genes were identified in this study, and the functional relevance of these genes was validated by gene expression analysis and network enrichment analysis. The authors may discuss the potential clinical application of these findings.

Additional comments

In this paper, the authors presented a study to use exome sequencing for investigating the genetic basis underlying high myopia. The authors proposed a phenotype-driven approach for filtering and analyzing genetic variants, followed by expression analysis and network enrichment analysis, which provided relatively strong evidence for the functional roles of the uncovered disease genes. The experimental design was reasonable and the results were informative and valuable. Therefore, this paper was believed to make some contributions to the research community and should be accepted.

Reviewer 2 ·

Basic reporting

The manuscript is well written and clear. This is an interesting genetic study in the clinical setting. The following are my comments:
1. Line 55, "disease-causing genes": in genetic studies, people usually use the "disease risk genes" or "disease-associated genes". Finding causal-relationship is challenging and usually needs detailed downstream functional experiments performed in addition to the genetic study. If the genes are merely associated with the disease by previous studies, I would suggest changing the wording to "disease risk genes" or "disease-associated genes".
2. Preprocessing the data: line 129, RPKM was transformed by log2. People usually use log2(RPKM + 1) to avoid negative values. Without adding the pseudo count, the numerical behavior for RPKM close to 0 will be problematic. Please use log2(RPKM + 1) and redo the analysis and figure 2.
3. Line 167, how was the 709 genes assembled? This should be described in detail in the methods section.
4. Details for some statistical analysis should be described. Line 174, what statistical test is performed to obtain the p-value? This should be described in detail in the methods section.
5. Line 147, "a total of 20 potential pathogenic (3 splicing, 1 frame-shift and 16 deleterious missense) heterozygous variants in 12 genes were identified in 16 patients". In line 116, "Pathogenicity of missense mutations was assumed if predicted pathogenic by at least 6 out of 8 computational methods". What happens if the criterion is relaxed (say 5 out of 8, 4 out of 8, ...)? How does that change the number of selected variants and downstream bioinformatics analysis?

Experimental design

Please see basic reporting

Validity of the findings

Please see basic reporting

Additional comments

Please see basic reporting

Reviewer 3 ·

Basic reporting

In the manuscript, Wan et al performed whole exome sequencing analysis for 20 individuals with myopia to identify pathogenic gene variants. This resulted in the identification of 20 potential disease-associated variants from 18 out of the 20 patients. Following that bioinformatics analyses were performed to investigate the relevant functional subnetworks and pathways enriched for genes carrying these variants. Overall, the manuscript is clearly written, but some revisions are required before the manuscript can be published.

Experimental design

(1) Causality. Limited by the experiment design and small sample size, this study cannot provide strong evidences that support the identified rare variants are causal. It is also difficult to calculate statistical significance of association of these variants with the disease. As shown all variants except one was identified only once (I assume the variant indented twice are from patients of the same family). Thus, it would be needed to discuss these limitations in the “Discussion”.

(2) Two of the patients are from the same family. It will be interesting to provide more information about their variants. I assume that these two patients are R0027 and R0029. Do the majority of variants identified shared by them? Genetically related individuals tend to share variants, but may inherit a different set of genetic variants. However, the variants causing the disease should be shared.

Validity of the findings

(1) Enrichment analysis. It is not clear what genes are used as background in the network and pathway enrichment analyses. Since these myopia-associated genes were selected by following a filtering protocol that focus on eye-disease related genes. It will certainly result in enrichment of related pathways and GO terms, if the background was not carefully selected (e.g., use all genes as the background for enrichment analysis). Thus the bioinformatics analyses reported in this study may not be valid.


(2) In the discussion, the authors described that many of the genes identified in this study have also been identified in previous studies. But it is not clear whether some of these 20 variants have also been reported in previous myopia sequencing studies.

Additional comments

Minor comments:
(1) In the Abstract, the meaning of “Our phenotype driven filtering method yielded much higher diagnostic rate than previous WES studies” is not clear. It should be removed from the abstract.


(2) Some sentences in the manuscript need to rephrased to avoid confusion. For example, in Line 100 “Variant calling was performed by a consensus call of at least 2 methods out of 4 haplotype-based calling algorithms”.

(3) Line 174, P< < 0.001 should be P<0.001.

---

## Round 0.2 · accepted · Accept

· Academic Editor

Accept

The reviewers agree that your revision has addressed all the comments to a reasonable extent. I suggest its acceptance.

# Reviewer 2 ·

Basic reporting

The revision has addressed all my comments to a reasonable extent.

Experimental design

The revision has addressed all my comments to a reasonable extent.

Validity of the findings

The revision has addressed all my comments to a reasonable extent.

Additional comments

The revision has addressed all my comments to a reasonable extent.

Reviewer 3 ·

Basic reporting

All my comments have been addressed. I thank the authors for carefully revising the manuscript.

Experimental design

None.

Validity of the findings

None.

Additional comments

All my comments have been addressed. I thank the authors for carefully revising the manuscript.